# Roles of electrons and ions in formation of the current in mirror mode structures in the terrestrial plasma sheet: MMS observations

Guoqiang Wang[1, 2], Tielong Zhang[1, 3], Mingyu Wu[1], Daniel Schmid[3], Yufei Hao[4],

Martin Volwerk[3]

[1]Institute of Space Science and Applied Technology, Harbin Institute of Technology, Shenzhen, China
[2]Key Laboratory of Lunar and Deep Space Exploration, Chinese Academy of Sciences, Beijing, China
[3]Space Research Institute, Austrian Academy of Sciences, Graz, Austria
[4]Key Laboratory of Planetary Sciences, Purple Mountain Observatory, Chinese Academy of Sciences, Nanjing, China

## Abstract

Mirror mode structures widely exist in various space plasma environments. Here, we investigate a train of mirror mode structures in the terrestrial plasma sheet on 11 August 2017 based on the Magnetospheric Multiscale mission. We find that bipolar current densities exist in the cross-section of two hole-like mirror mode structures, referred to as magnetic dips. The bipolar current density in the magnetic dip with a size of ~2.2 $\rho_i$ (the ion gyro radius) is mainly contributed by variations of the electron velocity, which is mainly formed by the magnetic gradient-curvature drift. For another magnetic dip with a size of ~6.6 $\rho_i$, the bipolar current density is mainly caused by an ion bipolar velocity, which can be explained by the collective behaviors of the ion drift motions. The current density inside the mirror dip contributes to the maintenance of the hole-like structure's stable. Our observations suggest that the electrons and ions play different roles in the formation of currents in magnetic dips with different sizes.

# 1 Introduction

Mirror modes are pressure-balanced and compressional magnetic structures (Hasegawa, 1969; Tsurutani et al., 2011; Wang et al., 2016; Zhang et al., 2018). They widely exist in many space plasma regions, such as solar wind (Zhang et al., 2008, 2009; Russell et al., 2009), planetary magnetosheath (Volwerk et al., 2008; Schmid et al., 2014), planetary magnetosphere (Vaivads et al., 2001; Rae et al., 2007), and comets (Glassmeier et al., 1993; Volwerk et al., 2016). These structures are believed to be generated by the mirror instability excited in the mirror unstable environment (Hasegawa, 1969; Southwood and Kivelson, 1993). The plasma perpendicular temperature anisotropy provides free energy to excite the mirror instability (Kivelson and Southwood, 1996). Once the mirror mode structures are generated, they will convected with the ambient flow since they are non-propagating relative to the ambient flow (Tsurutani et al., 2011). Due to gradients in the magnetic field and plasma density, the mirror mode structure may slowly propagate relative to the ambient plasma flow (Hasegawa, 1969, Pokhotelov et al., 2003). It is expected that they will stop to grow or decay when they move into the mirror stable region. Actually, they are reported to be able to survive in the mirror stable region in the solar wind and magnetosheath (Balikhin et al., 2009; Russell et al., 2009).

Mirror mode structures appear as not only quasi-periodic sinusoidal oscillations, but also local enhancements or decrease of the magnetic field intensity, referred to as magnetic peaks or dips (Tsurutani et al., 2011). Magnetic peaks can only exist in the mirror unstable environments, while magnetic dips are able to survive in the mirror stable region (Kuznetsov et al., 2007; Soucek et al., 2008). The typical scales of the mirror mode structures are 10s $\rho_i$ in the magnetosheath (Tsurutani et al., 1982; Horbury and Lucek, 2009), where $\rho_i$ is the ion gyro radius. Based on observations of the four Cluster satellites, the longest scales of the mirror mode structures in the magnetosheath is found to be 2 – 6 times length of their shortest scales, and their shapes are approximately cigar-like (Horbury and Lucek, 2009). By contrast, magnetic dips with

a scale less than 1 $\rho_i$ also exist in the magnetosheath as well as in the plasma sheet, and
electron vortices are found inside the structure (Ge et al., 2011; Huang et al., 2017, 2018,
2019; Yao et al., 2017).

In the terrestrial plasma sheet, there also exist mirror mode structures with several
ion gyro radii (Vaivads et al., 2001; Zieger et al., 2011; Li et al., 2014; Wang et al.,
2016). The earthward fast flows can result in a magnetic pileup in its leading area, and
the ion perpendicular temperature anisotropy in the pileup region is able to make the
local plasma conditions mirror-unstable to generate mirror mode structures (Zieger et
al., 2011). Mirror mode structures accompanied by electron dynamics and whistler
waves are also reported to occur during the dipolarization processes (Li et al., 2014;
Huang et al., 2018). Dipolarization fronts (DFs), characterized by a sharp enhancement
in $B_Z$ in GSM, are formed ahead of the earthward fast flows (Ge et al., 2012; Wu et al.,
2013; Schmid et al., 2016; Xiao et al., 2017). They play an important role in the energy
conversion, mass transport, particle accelerations and wave activities (Fu et al., 2012b;
Huang et al., 2012, 2015b). They are able to create a pressure pileup region ahead of
the DF when moving earthward (Schmid et al., 2011; Liu et al., 2013). Mirror mode
structures with a scale of ~4 $\rho_i$ are reported to occur in the pressure pileup region ahead
of a DF, and the mirror instability is suggested to be a potential mechanism to generate
these structures since local environments are mirror-unstable (Wang et al., 2016).
Within a mirror mode structure there should be an electric current driven by the
magnetic gradient and curvature drifts of the ions and/or electrons in order to sustain
their stability (Constantinescu, 2002).

In this study, we investigate a train of ion-scale mirror mode structures in the
terrestrial plasma sheet on 11 August 2017 using the Magnetospheric Multiscale (MMS)
mission data. Our aim is to figure out whether the main contributor to the current
density inside the ion-scale mirror mode structure is the electron or ion.

## 2   Observation

The MMS spacecraft consist of four identical satellites, which constitute a tetrahedron with inter-spacecraft distances of tens km (Burch et al., 2015). In the present study, we use the survey (a resolution of 16 Hz) magnetic field data obtained by the Fluxgate Magnetometer (Russell et al., 2016), and the survey (4.5 s) plasma data recorded by the Fast Plasma Instrument (Pollock et al., 2016).

### 2.1 Overview of a DF event

Figure 1 shows that $B_Z$ sharply increases ~8 nT within 7 seconds accompanied by a fast earthward flow with a maximum speed of ~397 km/s at ~20:38 UT on 11 August 2017. Also, the local ion beta, the ratio of the ion thermal pressure to the magnetic pressure is ~4, and the elevation angle ($\theta = \arctan\left(\frac{B_Z}{\sqrt{B_X^2+B_Y^2}}\right)$) changes ~50 ° with a maximum angle of 64 ° (not shown). These observations satisfy the criteria of the DF from Fu et al. (2012a), indicating that it is a DF event shown as the vertical dotted line at around 20:38 UT in Figure 1. At 20:40 UT, the MMS spacecraft are located near (-18, 14.6, 2) $R_E$ in GSM (Geocentric Solar Magnetospheric coordinates, used everywhere unless otherwise stated). The normal direction of the DF is (0.34, 0.82, -0.46) determined by the minimum variance analysis (MVA) (Sonnerup and Scheible, 1998) using the data in the interval between 20:37:33 and 20:37:42 UT. The ratio of the intermediate to minimum eigenvalues ($\lambda_2/\lambda_3$) is ~15, indicating that the estimated normal direction is reliable (Volwerk, 2006; Wang et al., 2014). The estimated normal direction suggests that the MMS spacecraft are located at the duskward side of the DF based on the semi-circle assumption of the DF (Huang et al., 2015a).

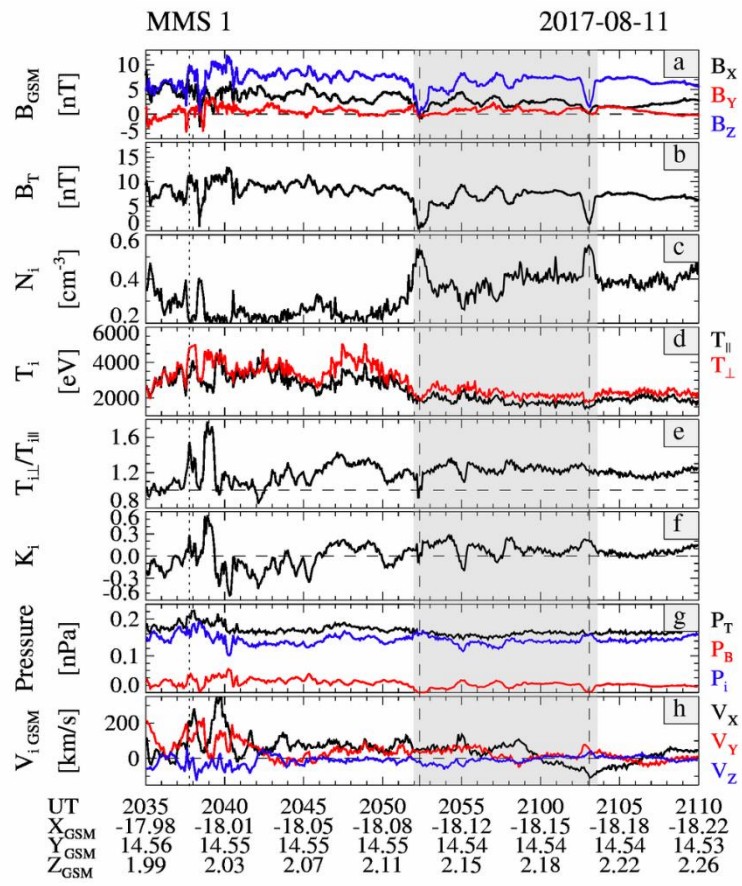


**Figure 1.** Observations of a DF event by MMS 1 on 11 August 2017. From top to bottom: three
components of the magnetic field in GSM (a), the total magnetic field (b), ion density (c), ion
perpendicular (red) and parallel (black) temperatures (d), ion perpendicular temperature anisotropy
(e), the threshold of the mirror instability (f), the magnetic, ion thermal and total pressures (g), and
three components of the ion velocity in GSM (h). The gray shadow indicates several compressional
structures. The vertical dotted line indicates the DF, and the dashed lines indicate the trough of two
hole-like structures.

Several quasi-periodic compressional magnetic oscillations with a period of ~2 min
are observed in the interval between 20:51 and 21:04 UT shown as the gray region in
Figure 1. The total magnetic field varies in anti-phase with the ion number density
during this interval. In addition, the total pressure, sum of the magnetic and ion thermal
pressures, is almost constant, indicating that they are pressure-balanced structures. The
threshold of the ion mirror instability $K_i$ is shown in Figure 1f, where $K = \frac{T_\perp}{T_\parallel} - 1 -$
$\frac{1}{\beta_\perp}$, and $T_\perp$, $T_\parallel$, and $\beta_\perp$ are perpendicular and parallel ion temperatures and
perpendicular ion beta, respectively (Southwood, and Kivelson, 1993). Local plasma
environments become mirror unstable and can excite ion mirror instabilities when $K_i >$
0. The maximum $K_i$ in each compressional structure reaches over 0.2, and it tends to
decrease to near or below 0 from the center of each structure to its edge. Before 20:51
UT or after 21:04 UT, $K_i$ is near or below 0, i.e. the background environment for these
structures is marginally mirror stable.

The above properties of the compressional structures indicate that they are likely to
be mirror mode structures (Tsurutani et al., 2011). Mirror mode structures are supposed
to be non-propagating structures relative to the ambient flow if there are no significant
gradients in the magnetic field and plasma density (Pokhotelov et al., 2003). Burst
magnetic field data (a resolution of 128 Hz) are available only between 20:51 and 20:54
UT, thus, we perform timing analysis (Harvey, 1998) to calculate the propagating
velocity of the hole-like structure between 20:51:55 and 20:52:56 UT to verify whether
these compressional structures are non-propagation. Figure 2A shows the positions of
the MMS spacecraft relative to MMS1 at 20:52 UT. The inter-spacecraft distances are
~13 to 21 km. Before performing the timing, the magnetic field data have been low-
pass filtered with a cutoff period of 30 s to reduce the effect of high frequency
fluctuations. Figure 2B shows the cross correlations between MMS1 and the three other
satellites by using $B_Z$. The maximum correlation coefficients are all almost 1 between
MMS1 and MMS2/3/4 with a lag time of -0.312 s, -0.164 s and -0.039 s, respectively.
The estimated velocity is (71.3, 11.7 °, -28 °) in spherical coordinates (r, θ, φ) transferred
from GSM coordinate system, where θ and φ are the longitude and latitude, respectively.
By contrast, the average ion velocity is (71.6, 37.8 °, -28.4 °) in this interval. Comparing
these two velocities, one can find that the compressional structures in Figure 1 are
approximately stationary, i.e. they are mirror mode structures.

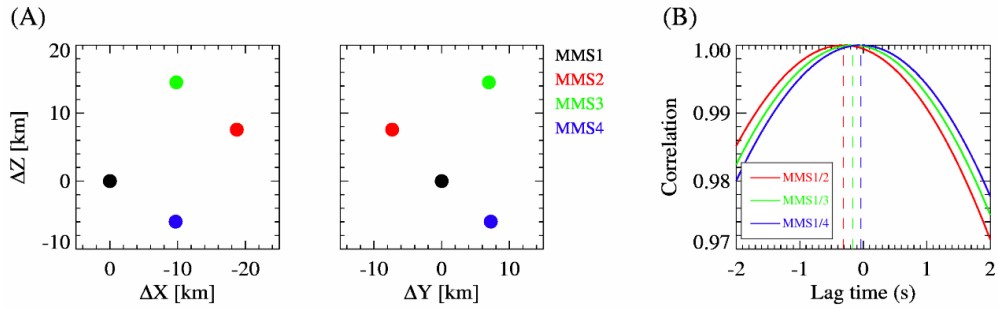


**Figure 2.** (A) Positions of the MMS spacecraft relative to MMS1 at 20:52 UT in the X-Z (left) and Y-Z (right) planes. (B) The cross correlations between MMS1 and the three other MMS satellites calculated by using $B_Z$ in the interval 20:52:55 – 20:52:56 UT.

The first and last mirror mode structures as the dashed lines shown in Figure 1 are hole-like, which are referred to as magnetic dips. We will focus on these two magnetic dips in the rest paper, and we mark them as MM1 (20:51:55 – 20:52:56 UT) and MM2 (21:02:26 – 21:03:34 UT).

**2.2 Plasma properties in MM1**

To further look at the plasma properties in the magnetic dips, we transform the ion and electron velocities as well as the magnetic field and current density into the principal axis (LMN) coordinate system as shown in Figure 3. The principal axes vectors are calculated by MVA using the magnetic field data obtained from MMS1 in the interval between 20:51:55 and 20:52:56 UT. To reduce the effect of the high frequency fluctuations, the magnetic field data have been low-pass filtered with a cutoff period of 30 s before performing the MVA analysis. The **L**, **M** and **N** directions are (0.46, 0.27, 0.85), (0.28, 0.86, -0.42) and (-0.84, 0.43, 0.32) in GSM, respectively. The eigenvalue ratio $\lambda_2/\lambda_3$ is ~9.

Figure 3 shows that $B_L$ is dominant while $B_M$ and $B_N$ vary around 0. The angles between the average magnetic field in this interval and the **L**, **M** and **N** directions are ~18°, 108° and 87°, respectively. It indicates that the cross-section of MM1 is approximately parallel to the M-N plane, and is approximately perpendicular to the ambient magnetic field. The N direction is supposed to be parallel to the above

estimated velocity by timing, however, the angle between these two directions is ~37 °.
The MVA technique can be effected by waves or noises superimposed on the
discontinuity surface (Lepping and Behannon, 1980; Schmid et al., 2019), while the
inter-spacecraft distances and configuration of the MMS spacecraft can effect on the
accuracy of calculation (Harvey, 1998), which might a possible explanation for the
large difference between the two estimated normal directions. The ion velocity is
mainly in the M-N plane during the whole interval, and there are no significant changes
in both $V_{iM}$ and $V_{iN}$. By contrast, the N component of the electron velocity $V_{eN}$ shows
a bipolar variation with an amplitude of ~40 km/s. To reduce the effect of the high
frequency noise, the electron data have been smoothed within a 30-second window in
Figure 3 as well as in Figure 4. Interestingly, an enhancement (a decrease) of $V_{eN}$ occurs
in the left (right) side of MM1, i.e. a bipolar feature appears in $V_{eN}$.

The current density in Figure 3 is calculated by the curlometer technique (Dunlop et
al., 2002) using the magnetic field data low-pass filtered with a cutoff period of 30 s.
The current density can be regarded as reliable when the ratio $|\nabla \cdot B|/|\nabla \times B|$ is less than
0.2 (e.g., Wang et al., 2017, 2019). The N component of the current density $j_N$ shows a
bipolar variation similar to $V_{eN}$ with an opposite trend of change. The correlation
coefficient between $j_N$ and $V_{eN}$ inside MM1 is -0.97. By comparing the variations in the
ion and electron velocities, one can note that the bipolar current density inside MM1 is
mainly associated with the electron velocity. The peak and trough of the bipolar $V_{eN}$
tend to occur near the maximum gradient of $B_L$, while there is no significant change in
$P_{e\perp}$.

Since the magnetic dips are stationary in the ambient flow, we can estimate their scale
in the cross-section by
$$\sqrt{\left(\int_{t_1}^{t_2} V_M dt\right)^2 + \left(\int_{t_1}^{t_2} V_N dt\right)^2}$$

where $V_M$ and $V_N$ are the M and N components of the ion velocity, $t_1$ and $t_2$ are the start
and end times of each magnetic dip. The scale of MM1 is estimated to be ~4.1 $\times$ $10^3$
km, or ~2.2 $\rho_i$, where $\rho_i$ is the local ion gyro radius calculated by the average ion
perpendicular temperature and the average $B_T$ in MM1 between 20:51:55 – 20:52:56
UT. Since the spacecraft may not cross the center of the magnetic dip, the estimated
scale is the lower limit.

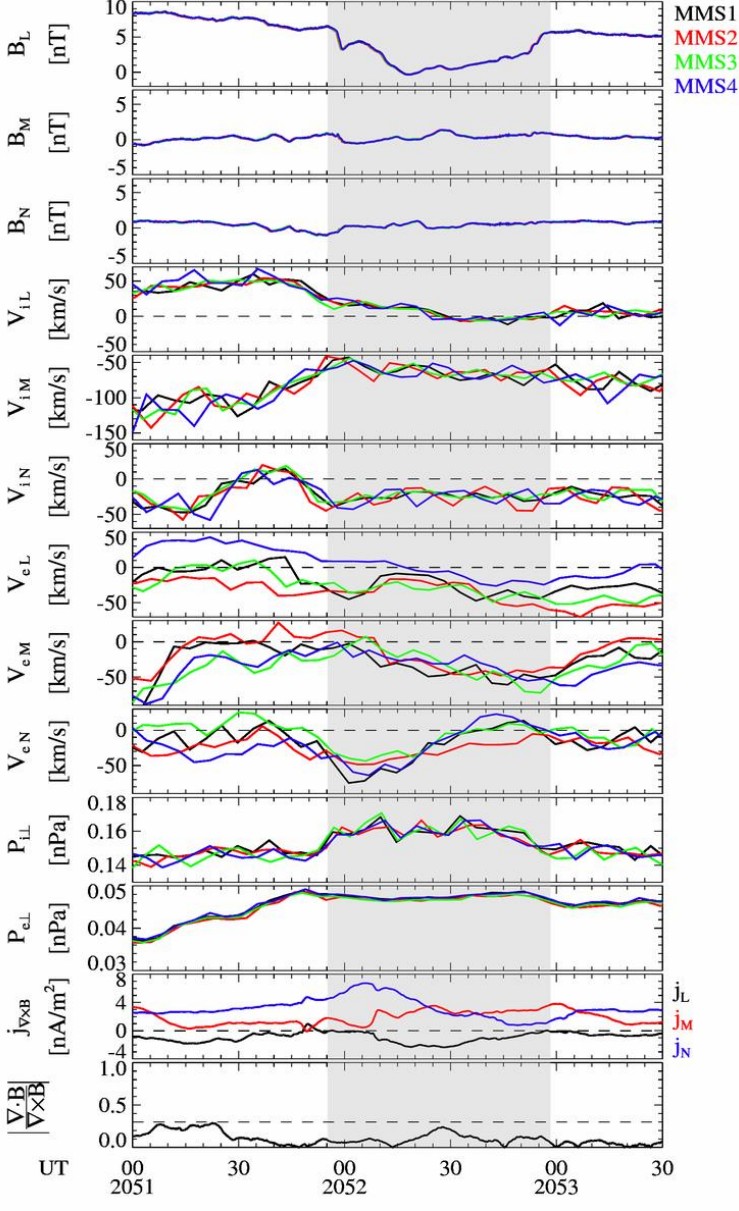


**Figure 3.** From top to bottom: three components of the magnetic field, ion and electron velocities
in LMN, the ion and electron perpendicular thermal pressures, the current density in LMN and the
ratio of $|\nabla \cdot B|/|\nabla \times B|$ between 20:51 and 20:53:30 UT. The black, red, green and blue colors
indicate data obtained from MMS1, MMS2, MMS3 and MMS4, respectively. The current density
is calculated by the curlometer technique. The gray region indicates the interval of the magnetic dip.

**2.2 Plasma properties in MM2**

Figure 4 shows the magnetic field, ion velocity, electron velocity and current density in LMN between 21:01 and 21:05 UT. The magnetic field data between 21:02:26 and 21:03:34 UT are used to calculate the principal axes vectors by MVA. The ratio $\lambda_2/\lambda_3$ is ~6, and the **L**, **M** and **N** directions are (0.26, 0.1, 0.96), (-0.44, 0.89, 0.02) and (-0.86, -0.43, 0.28), respectively. The angles between the average magnetic field in this interval and the **L**, **M** and **N** directions are ~1.5°, 89° and 89°, respectively. $B_L$ is dominant during the whole interval, while $B_M$ and $B_N$ are very small. Thus, the cross-section of MM2 is also approximately parallel to the M-N plane, and almost perpendicular to the ambient magnetic field. No large-amplitude fluctuations appear in MM2 compared to MM1. The ion velocity $V_{iM}$ and $V_{iN}$ are dominant, while $V_{iL}$ varies around 0. Interestingly, a bipolar feature in $V_{iN}$ with a variation up to 80 km/s (peak minus trough) can be distinctly found inside the dip, while $V_{iM}$ tends to increase compared to the ambient flow. $V_{iN}$ is smaller (larger) than the ambient value in the left (right) side of the dip. The peak and trough of the bipolar $V_{iN}$ appear when there are significant gradients in the magnetic field and the ion perpendicular thermal pressures. It indicates that the bipolar $V_{iN}$ could be associated with the magnetic gradient and diamagnetic drifts. The length of MM2 in the cross-section is estimated to be ~6.4 $\times$ $10^3$ km, or ~6.6 $\rho_i$.

235

The current density in Figure 4 is also determined by the curlometer technique. Before performing the curlometer analysis, the magnetic field data have been low-pass filtered with a cutoff period of 20 seconds to reduce the effect of the high-frequency fluctuations. One can find that $j_N$ shows a similar bipolar feature to $V_{iN}$. The correlation coefficient between $V_{iN}$ and $j_N$ is 0.92 in the whole interval of MM2, indicating that both parameters have a strong relation. The peak minus the trough of $j_N$ during MM2 is ~5.6 nA/m$^2$. By contrast, $j_L$ and $j_M$ have no such a clear bipolar feature. The electron velocities show variations with periods larger than 1 minute, but no clear bipolar feature appears in any component of the electron velocity during MM2, indicating that the

bipolar $j_N$ is mainly determined by $V_{iN}$.

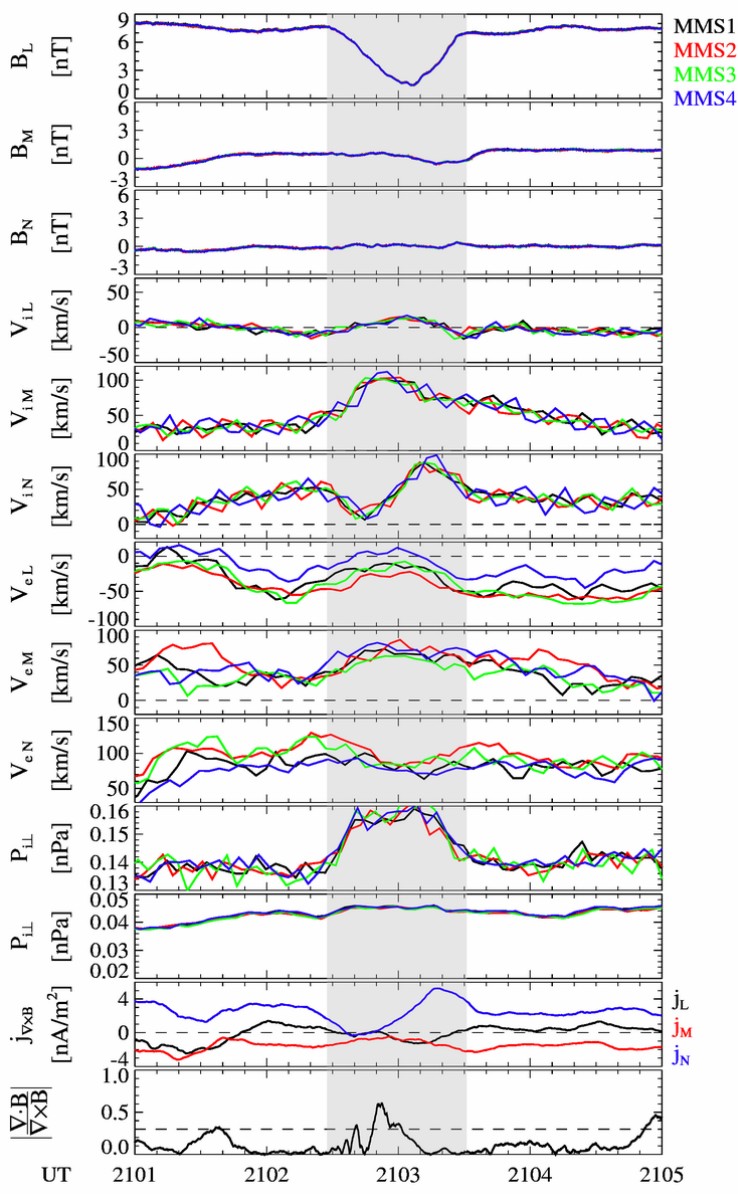

**Figure 4.** From top to bottom: Three components of the magnetic field, ion and electron velocities
in LMN, the ion and electron perpendicular thermal pressures, the current density in LMN and the
ratio of $|\nabla \cdot B|/|\nabla \times B|$ between 21:01 and 21:05 UT. The black, red, green and blue colors indicate
data obtained from MMS1, MMS2, MMS3 and MMS4, respectively. The current density is
calculated by the curlometer technique. The gray region indicates the interval of the magnetic dip.

253        To look at the variations of the ion flow in MM2, we assume that the ion velocity
observed during MM2 consists of $V_{i\_a}$ and $V_{i\_md}$, where $V_{i\_a}$ is the ambient ion velocity,
and $V_{i\_md}$ is the ion velocity inside MM2 relative to the ambient flow. The average
velocity 30 seconds before and after MM2 is selected to be regarded as $V_{i\_a}$ with a value
of (-2.6, 51.4, 33.4) km/s in LMN. Figure 5 shows the deflection of $\mathbf{V_{i\_md}}$ in the M-N
plane. The arrows indicate the direction of the ion velocity, and their lengths indicate
the magnitude of $\mathbf{V_{i\_md}}$ in the M-N plane. The direction of $\mathbf{V_{i\_md}}$ gradually changes from
around -60° to 50° in the M-N plane. Also, the strength of $\mathbf{V_{i\_md}}$ in this plane gradually
increases and then decreases from the left side of the magnetic dip to the right side. In
addition, the N component of $\mathbf{V_{i\_md}}$ changes from negative to positive at just around the
center of the structure.

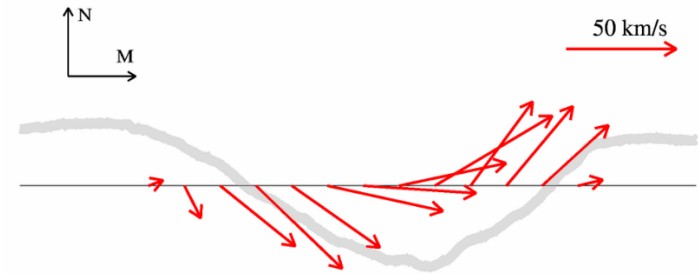


**Figure 5.** Ion velocities $\mathbf{V_{i\_md}}$ in the M-N plane during MM2. The arrows indicate the direction of
the ion velocities, and their lengths indicate the amplitude of the ion velocities. The gray line
indicates the total magnetic field of MM2.

## 270 3 Discussion

Since mirror mode structures are stationary in the ambient flow, we can estimate the
distance of the structures relative to the DF in the Y direction using the average $V_Y$ ~30
km/s during the structures. Thus, they are likely to occur dawnside of the MMS
spacecraft with a distance of ~4 $R_E$ in the Y direction when the spacecraft are crossing
the DF at around 20:38 UT. Compared this distance with the typical size of the DF (~3
$R_E$) (Huang et al., 2015a) and the size of the magnetic dips in Figure 1, the mirror mode
structures might come from the dawnside flank of the DF. Since the DF is considered
to be a tangential discontinuity (Schmid et al., 2019) which pushes the background
plasma to its flanks (Fu et al., 2012a, 2012b; Liu et al., 2013; Birn et al., 2015), the
plasma near the flank is expected to come from the pressure pileup region ahead of DFs.
In addition, mirror mode structures have been reported to be potentially generated in
such a pressure pileup region (Zieger et al., 2011; Wang et al., 2016). Thus, the mirror
mode structures in Figure 1 might originate from the pressure pileup region ahead of
the DF.

Based on Ampère's law, there should exist a current in the magnetic dip to sustain
the structure's stability (see Constantinescu, 2002). Figure 3 and 4 shows that a bipolar
current density is observed in both MM1 and MM2. $B_L$ changes ~5 nT in MM1 between
20:52:30 and 20:52:56 UT, and half of the estimated length of MM1 is $2.05 \times 10^3$
km in the cross-section. Assuming that $B_M$ and $B_N$ are 0, and $B_L$ changes just along the
trajectory of MMS, a current density $j_B$ with a value of ~2 nA/m$^2$ in the cross-section is
necessary to be self-consistent with the magnetic field depression. The amplitude of the
bipolar $j_N$ is ~2 nA/m$^2$ between 20:52:30 and 20:52:56 UT, almost equal to $j_B$, indicating
that MM1 is a stable structure (Constantinescu, 2002). Similarly, MM2 is also a stable
structure.

Significant changes can be found in electron velocities in MM1, while the three
components of the ion velocity are almost constant. Therefore, the current density in
MM1 is mainly contributed by electrons. The amplitude of the bipolar electron velocity
in $V_{eN}$ is ~40 km/s (see Figure 3). Three kind of the electron drift motions are expected
to create the current density, i.e. the magnetic gradient drift, the magnetic curvature
drift and the diamagnetic drift. The electron perpendicular thermal pressure $P_{e\perp}$
changes ~0.002 nPa in MM1, the average electron number density is ~0.4 cm$^{-3}$, and the
average total magnetic field is ~3 nT. Consequently, the estimated electron diamagnetic
drift velocity is ~4 km/s, much smaller than the amplitude of the bipolar $V_{eN}$. The peak
of the bipolar $V_{eN}$ occurs in the time interval between 20:52:40 and 20:52:50 UT, during
which there are no significant magnetic field fluctuations. We select this time interval
to estimate the velocities of the magnetic gradient and curvature drifts. The total
magnetic field changes ~1.1 nT, and the median total magnetic field is ~2.2 nT in this
interval. The median electron perpendicular and parallel temperatures are ~680 eV and
650 eV. The length scale of MM1 is ~4.1 $\times 10^3$ km in the M-N plane and its duration
is ~61 s, thus the length for the time interval between 20:52:40 and 20:52:50 UT is ~680
km. Using the data from all four MMS satellites, we can determine the curvature of
MM1 by
$$\rho_c = B^{-2} B_i \nabla_i B_j - B^{-4} B_j B_i B_l \nabla_i B_l$$
where the indices i, j and l indicates the three components of the magnetic field, and B
= |**B**| (Shen et al., 2003). The curvature radius $R_C$ is $1/\rho_c$. Before performing the
calculation, the magnetic field data have been low-pass filtered with a cutoff period of
1 second to reduce the effect of the high-frequency noise. The median $R_C$ in this interval
is 1.1 $\times 10^3$ km. Thus, the velocities of the electron magnetic gradient and curvature
drifts are ~209 km/s and 262 km/s, respectively. Since the magnetic curvature drift in
MM1 is in the opposite direction of the magnetic gradient drift., thus the collective
velocity of these two velocities are ~53 km/s, which is close to the amplitude of the
bipolar $V_{eN}$. It suggests that the bipolar electron velocity in MM1 is mainly formed by
the electron magnetic gradient and curvature drifts.

327        The size of MM1 is ~2.2 $\rho_i$, and its central magnetic field strength is almost 0. Thus,

the ion gyro radius is expected to significantly change within one orbit, and ions would
randomly jump between neighboring magnetic dips. These ions are referred to as
chaotic particles (B üchner and Zelenyi, 1989), which could be one reason why ions do
not seem to contribute to the formation of the current in MM1.

333        No significant changes occur in the electron velocity in MM2, thus the bipolar

current density is mainly contributed by the variations of the ion velocity (see Figure
4). The size of MM2 is ~6.6 $\rho_i$, larger than that of MM1. The trough of the bipolar $V_{iN}$
is observed at around 21:02:45 UT, meanwhile, $V_{iM}$ increases ~50 km/s compared to
the ambient flow on the left side of MM2. The amplitude of the bipolar $V_{iN}$ is ~50 km/s,
thus, the ion velocity inside MM2 ~70 km/s relative to the ambient ion flow. The ion
perpendicular thermal pressure tends to be larger from the edge of MM2 towards its
center (see Figure 4), therefore, an ion diamagnetic drift is expected to be formed
(Baumjohann and Treumann, 1997). We use the data in the time interval between
21:02:30 and 21:02:50 UT to estimate the ion thermal pressure and magnetic gradients.
Also, the average ion perpendicular and parallel temperatures, average total magnetic
field and average curvature radius in this interval are used to estimate the velocities of
the ion drift motions. Consequently, the velocities of the ion diamagnetic, magnetic
gradient and curvature drift motions are ~17 km/s, 33 km/s and 79 km/s, respectively.
By contrast, the velocities of the electron diamagnetic, magnetic gradient and curvature
drifts are ~5 km/s, 14 km/s and 36 km/s. Since the ion diamagnetic and magnetic
curvature drifts move almost in the same direction in the M-N plane, while the ion
magnetic gradient drift moves in the opposite direction. Thus, the collective drift
velocity is ~63 km/s, very close to the ion velocity inside MM2 with a speed of 70 km/s.
Thus, one can expect that the bipolar $V_{iN}$ in Figure 4 is the collective behaviors of the
ion drift motions in MM2.

Except for the bipolar $V_{iN}$, there is an enhancement of $V_{iM}$ in MM2. To figure out
the variations of $V_{iM}$ and $V_{iN}$ in MM2, we analyze the possible trajectory of the MMS
spacecraft crossing MM2. Mirror mode structures in the magnetosheath are found to be
cigar-like structures instead of sheets or tubes (Constantinescu et al., 2003; Horbury
and Lucek, 2009). To simplify our analysis, we assume that the cross-section of MM2
is a circle. To be self-consistent with the magnetic field depression, the ion flow as well
as the current is supposed to be clockwise as the black arrows shown in Figure 6. Based
on the normal directions of the both half sides of the structure along the spacecraft
trajectory and the ambient flow direction, we can get the possible trajectory of the MMS
spacecraft in the M-N plane. We calculate the normal directions of the two sides of
MM2 by MVA, and the values are (0.03, 0.79, 0.61) and (-0.05, -0.65 0.76) in LMN
for the intervals 21:02:30 – 21:03 and 21:03:10 – 21:03:25 UT, respectively. The ratios
of the intermediate to minimum eigenvalues $\lambda_2/\lambda_3$ are 6.4 and 8.5, respectively. The
normal directions are almost orthogonal to each other, thus, the maximum length of
MM2 in the cross-section could be 1.4 times the estimated length (6.6 $\rho_i$) based on the
assumption of a circle. The velocity of the ambient ion flow is (-2.6, 51.4, 33.4) km/s
in LMN. Thus, a possible trajectory of MMS in the M-N plane can be drawn based on
the ambient flow and the above normal directions as the red arrow shown in Figure 6.
Since the inter-spacecraft distances are very small compared to the scale of MM2, only
the possible trajectory of MM1 is shown in Figure 6. Along the trajectory, $V_{iN}$ changes
from negative to positive from one to another side of MM2, while $V_{iM}$ is positive, which
is in agreement with the deflection of the ion flow shown in Figure 5. Thus, the
variations of $V_{iM}$ and $V_{iN}$ are consistent with the prediction of the ion vortex in the
cross-section. Such a ring-like flow might play an important role in the evolution of the
mirror mode structure or maintaining the stability of the magnetic dip.

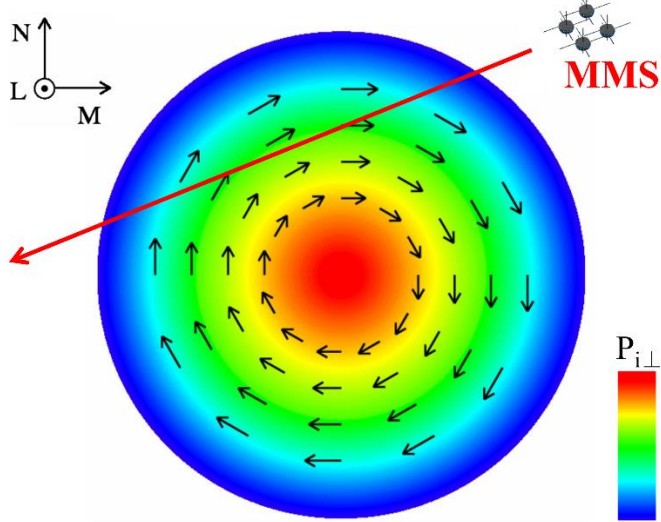


**Figure 6.** Schematic of MMS1 crossing the magnetic dip in the M-N plane. The colors changing
from center (red) of the magnetic dip to its edge (blue) indicate the decrease of the ion perpendicular
thermal pressure as shown by the color bar. The back arrows in the magnetic dip indicate the
direction of the ion velocity. The red arrow indicates a possible trajectory of MMS1.

**4   Summary**
We have studied the ion-scale mirror mode structures in the plasma sheet on 11
August 2017. We find that a bipolar current density in the magnetic dip with a size of
~2.2 $\rho_i$ is mainly contributed by an electron bipolar velocity in the cross-section. The
electron bipolar velocity mainly results from the magnetic gradient and curvature drifts.
The chaotic motion of ions might be one significant reason that ions have almost no
contribution to the formation of the bipolar current in this magnetic dip. For another
magnetic dip with a size of 6.6 $\rho_i$, the bipolar current is mainly contributed by the ion
bipolar velocity, which can be explained by the collective behavior of the ion drift
motions. And the variations of the ion velocity in the cross-section suggest the potential
existence of the ion vortex. We suggest that the scale as well as the magnetic geometry
of the magnetic dip is significant to determine the roles of electrons and ions in the
formation of the current inside the dip.

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

## Author contribution

Guoqiang Wang and Tielong Zhang designed the main idea of this study, and the data analysis was mainly performed by Guoqiang Wang. Guoqiang Wang prepared the manuscript with contributions from all co-authors.

## Acknowledgements

This work in China was supported by NSFC grants 41804157, 41774171, 41974205, 41774167, and 41904156. The authors also acknowledge the financial supported by the grant from Key Laboratory of Lunar and Deep Space Exploration, CAS, Shenzhen Science and Technology Research Program (JCYJ20180306171918617), and the 111 project [B18017]. We acknowledge the data from the NASA MMS mission. We also acknowledge MMS project FGM and FPI teams. The data of the MMS spacecraft are publicly available at https://lasp.colorado.edu/mms/sdc/public/.

## Code/Data availability

The data of the MMS spacecraft are publicly available at https://lasp.colorado.edu/mms/sdc/public/.

## Competing interests

The authors declare that they have no conflict of interest.