# Peer review of "Roles of electrons and ions in formation of the current in mirror mode structures in the terrestrial plasma sheet: MMS observations"

_Annales Geophysicae, 2019_

## Referee Comment (RC1) · Anonymous Referee #1 · 13 Dec 2019

**1 General remarks**

The submitted manuscript investigates the electric current distribution within two magnetic dips identified as mirror mode structures in the terrestrial plasma sheet. As these are quasi-stationary magnetic field structures in the plasma frame, they must be supported by electric currents. According to the Authors, the currents are carried preponderantly by either electrons or ions, depending on the scale of the structure. To my knowledge this is the first experimental study of these current systems, therefore the manuscript can add a valuable contribution to our current understanding of the mirror modes. There are however a number of issues which should be addressed before

publication.

Despite the availability of magnetic field and particle data from the four MMS space-craft forming a "tetrahedron with inter-spacecraft distances of tens km" – as mentioned in page 2, line 79 of the manuscript, little advantage of the multi-point measurements is taken by the Authors. As far as I can tell, the multi-point capabilities of the MMS fleet were only used to determine the spacecraft-frame velocities of the detected compressional fluctuations (page 5-6, lines 113-120). Everywhere else, only single spacecraft data seems to be used. I am aware that the tetrahedron configuration might not be appropriate for some multi-point techniques, such as the curlometer, or that the characteristic size of the tetrahedron might not be ideal for the scale of the investigated structures. Nevertheless, the Authors should either use the measurements from all spacecraft or clearly explain why some of the data is excluded from the analysis. There is only a brief remark in this direction in the manuscript, stating that the interspacecraft distances are to small to allow an estimation of the magnetic field curvature (page 12, lines 270-272).

Even when essentially single spacecraft data are used (e.g. determining the principal coordinate system, scales of the structures, instability condition, current densities, pressures, particle velocities), reference should be made to all four MMS spacecraft, differences between spacecraft discussed, and when possible mean values used. In particular, figures 2 and 3 should include all spacecraft.

The text should be better structured and the language should be revised throughout the manuscript.

**2  Specific comments**

• page 2, line 37-39

Due to gradients in the magnetic field and plasma density, the mirror mode waves may slowly propagate relative to the ambient plasma flow (Hasegawa 1969, Pokhotelov JGRA 2003).

• page 5-6, line 115-120

More details about the timing method used to estimate the velocity of the compressional oscillations should be given. What are the time delays, accuracy? Tetrahedron size, elongation and planarity should be discussed. Is the determined speed the phase velocity in the spacecraft frame? (i.e. planar wave fronts orthogonal to the determined velocity vector are assumed? – if yes, then the direction of the determined velocity vector should be compared with the minimum variance direction determined on page 7, line 153. They should agree.). Since the Authors refer to the oscillations between 20:51 and 21:04 (page 5, line 112) why only the interval [20:51:55, 20:53], corresponding to the later identified (page 7, Table 1) MM1 structure, is used? To ease the interpretation and comparison between the determined phase velocity vector and the mean plasma flow velocity, spherical coordinates (magnitude, $\theta$, $\varphi$) should be used, and the angle between the two vectors should be given.

(Harvey 1998) does not appear in the manuscript references list. I assume it is Chapter 12 in the ISSI "Analysis Methods for Multi Spacecraft Data" book.

• page 6, line 127-135

The velocity used for estimating the scales (line 129) should be the one determined from timing analysis, not the plasma flow velocity. Since the two are not very different (line 118), this should not change much the results. Most probably the mirror mode structures have different sizes in different directions. For this study, the relevant size is the size in the direction orthogonal to the magnetic field. This size should be determined considering the angle between the mean magnetic field and the velocity vector determined from the timing analysis. Since the minimum variance direction – which

should be close to the velocity direction – seems to be orthogonal to the mean magnetic field (figures 2 and 3), I expect that the sizes estimated in the manuscript are not far from the sizes in the orthogonal to the mean field direction. However, if the structures are not crossed through their centers – e.g. a path similar to the one shown in Figure 5 –, then the estimated sizes are only lower limits.

On lines 131-132 I assume the Authors meant "average ion perpendicular temperature".

• page 7, Table 1

"$\rho_i$" should read "Scale ($\rho_i$)".

• page 7, lines 147-159

After line 147 the manuscript concentrates only on two magnetic dips (MM1 and MM5). To help readability, this should be clearly stated. The first structure (MM1) is analysed in this paragraph and in the next one (up to line 181), while MM5 is analysed in the remaining of the section. Dividing the text in subsections would improve readability.

In this context, the maximum variance direction – which for magnetic mirrors should be aligned with the mean magnetic field – is the important direction. Therefore the ratio between the maximum and the intermediate eigenvalues is relevant. The angles between the mean magnetic field and the determined $\mathbf{L}, \mathbf{M}$ and $\mathbf{N}$ directions should be given.

The current density should be computed also using the curlometer, or the Authors should explain why this technique cannot be applied.

Same comments apply for the MM5 on the next page.

• Figures 2 and 3

Figures 2 and 3 should show the orthogonal pressures of both ions and electrons. Are the ion velocities and the electron pressure in Figure 2 smoothed?

• page 7-8, lines 161-174

A more quantitative approach to determine which species (ions or electrons) contribute mostly to the electrical current is desirable. The Authors might e.g. compute the correlation between the electrical current and the ion and electron velocities.

• page 11, lines 240-242

Please state the assumptions made for estimating the current density $j_B$.

• page 11, lines 251-255

There is no reference to chaotic particles in (Constantinescu 2002). Perhaps the Authors refer to another paper?

• page 12-13, lines 285-295

An estimation of the gradient drift velocities for electrons and ions (similar with the estimation done in the previous paragraph for MM1), as well as an estimation of the electron diamagnetic drift should be given.

• page 13, lines 301-309

The normal directions (line 305) are almost orthogonal to each other. Knowing the estimated size between the entry and exit points, $d$, one can derive the transversal size of the structure as illustrated in Figure 5 (about $1.4d$). Why is the MMS trajectory a curved line? Does the assumed relative motion of the magnetic structure change so much during the crossing time?

---

## Referee Comment (RC2) · Anonymous Referee #2 · 18 Dec 2019

Wang et al. investigated the roles of electrons and ions in the formation of the current in the mirror mode structures in the plasma sheet using by the MMS observations. They found that the electrons and ions play a different role in the different sizes of mirror mode structures: the current carriers are mainly the electrons in small size mirror mode by magnetic gradient-curvature drift, and the ions in large size mirror mode by the ion diamagnetic drift. This study sheds new light on formation of currents in the mirror modes, and is worthy of publication in AG after moderate reversion.

In the discussion section: MMS consists of four identical spacecraft, and could provide the simultaneous measurements of four points. Why the authors use the plasma

measurements and magnetic field to estimate the time series of magnetic gradient curvature drift, electron diamagnetic drift, ion diamagnetic drift, and other terms. I think this is useful to estimate these different terms and then compare them.

Line 18 It would be better to replace "data " to "instruments", or remove "data".

Line 47-49 Actually the sizes of magnetic holes can be less than ion cyclotron radius in the magnetosheath. Such magnetic holes, named as kinetic-size magnetic hole or electron vortex magnetic hole, are widely observed using by MMS (doi:10.3847/1538-4357/ab0f2f, doi:10.1002/2017JA024415, doi.org/10.3847/1538-4357/aac831, doi:10.1002/2016JA023858).

Line 54-56 The small-size magnetic holes, below one ion cyclotron radius, are also detected in the plasma sheet (doi.org/10.3847/1538-4357/ab0f2f). These magnetic holes are always accompanied with electron scale instabilities, such as whistler waves.

Line 61-63 Dipolarization fronts are widely investigate in many literatures (doi:Âǎ10.1002/2015JA021083, doi:10.1029/2012GL051784, doi:10.5194/angeo-30-97-2012), and they play an important role in the energy conversion, mass transport, particle accelerations, and wave activities.

Line 72-75: I suggest the author give the motivation of this paper to help the readers to better understand their work.

Line 80-83 (Russell et al., 2014) should be corrected to (Russell et al., 2016) . If the author did not use the burst mode data in this paper, I suggest the authors remove the introduction about the resolution of burst mode in this part.

Line 115-117 Why are the data performed low-pass filtered before the timing analysis? I suggest the authors give some descriptions here.

Line 120 "tends to be larger" "increases" or "has a peak"?

Line 131: how to calculate the local ion gyro radius? Which time interval? Please give

the details in the text.

133-135 Why the range of the scale and the angle are inconsistent with those in table 1? The "rotation angle" is the "shear angle"?

Line 155: As the authors know, the separation of the four MMS spacecraft is very small. Thus, one can use the curlmeter method to estimate the current density basded on the magnetic field from four spacecraft. Why not the author use this method to calculate the current and compare with the current derived from the plasma measurements?

Line 192 JN should be corrected to JM.

Line 216: "Ion velocities" should be "Ion velocities Vi_md"?

Line 230-235 The pileup region usually exists behind the DFs, not ahead of the DFs, for example the definition of flux pileup region in the paper (doi:10.1029/2012JA018141). In addition, the mirror mode structures are observed after the detection of DF. Why the authors thought they originate from the pileup region ahead of the DF?

Line 243 Please indicate at which time "The amplitude of the bipolar jN in MM1 is ∼2nA/m2 ".

Line 247 It would be better to add "in MM5" after "electron velocity " Line 271-272 The authors can try to use the magnetic field from four MMS spacecraft to estimate the curvature radius of mirror modes.

Line 286-187 Did the authors ever calculate the magnetic gradient drift velocity in MM5? It would be necessary to compare the magnetic gradient drift velocity and the Vi_N.

Line 304-307 Did the authors compare the normal directions calculated by MVA and timing method to ensure the accuracy of the results.

Line 307-309 Please indicate which MMS? 1 or 2 or 3 or 4 after "trajectory of MMS". How to deduce the possible trajectory of MMS, please give details in the text.

Line 316 Please add a colorbar in Figure 5.

---

## Author Comment (AC1) · 21 Jan 2020

General remarks The submitted manuscript investigates the electric current distribution within two magnetic dips identified as mirror mode structures in the terrestrial plasma sheet. As these are quasi-stationary magnetic field structures in the plasma frame, they must be supported by electric currents. According to the Authors, the currents are carried preponderantly by either electrons or ions, depending on the scale of the structure. To my knowledge this is the first experimental study of these current systems, therefore the manuscript can add a valuable contribution to our current understanding of the mirror modes. There are however a number of issues which should be addressed

before publication.

Despite the availability of magnetic field and particle data from the four MMS spacecraft forming a "tetrahedron with inter-spacecraft distances of tens km" – as mentioned in page 2, line 79 of the manuscript, little advantage of the multi-point measurements is taken by the Authors. As far as I can tell, the multi-point capabilities of the MMS fleet were only used to determine the spacecraft-frame velocities of the detected compressional fluctuations (page 5-6, lines 113-120). Everywhere else, only single spacecraft data seems to be used. I am aware that the tetrahedron configuration might not be appropriate for some multi-point techniques, such as the curlometer, or that the characteristic size of the tetrahedron might not be ideal for the scale of the investigated structures. Nevertheless, the Authors should either use the measurements from all spacecraft or clearly explain why some of the data is excluded from the analysis. There is only a brief remark in this direction in the manuscript, stating that the interspacecraft distances are to small to allow an estimation of the magnetic field curvature (page 12, lines 270-272).

Even when essentially single spacecraft data are used (e.g. determining the principal coordinate system, scales of the structures, instability condition, current densities, pressures, particle velocities), reference should be made to all four MMS spacecraft, differences between spacecraft discussed, and when possible mean values used. In particular, figures 2 and 3 should include all spacecraft.

The text should be better structured and the language should be revised throughout the manuscript.

2 Specific comments Page 2, line 37-39 Due to gradients in the magnetic field and plasma density, the mirror mode waves may slowly propagate relative to the ambient plasma flow (Hasegawa 1969, Pokhotelov JGRA 2003).

Answer: Thanks for your nice suggestion. We have added this sentence in our revised manuscript.

Page 5-6, line 115-120 More details about the timing method used to estimate the velocity of the compressional oscillations should be given. What are the time delays, accuracy? Tetrahedron size, elongation and planarity should be discussed. Is the determined speed the phase velocity in the spacecraft frame? (i.e. planar wave fronts orthogonal to the determined velocity vector are assumed? if yes, then the direction of the determined velocity vector should be compared with the minimum variance direction determined on page 7, line 153. They should agree.). Since the Authors refer to the oscillations between 20:51 and 21:04 (page 5, line 112) why only the interval [20:51:55, 20:53], corresponding to the later identified (page 7, Table 1) MM1 structure, is used? To ease the interpretation and comparison between the determined phase velocity vector and the mean plasma flow velocity, spherical coordinates (magnitude, $\theta$, ÏT) should be used, and the angle between the two vectors should be given.

Answer: Thanks for your nice comments and suggestions. Burst magnetic field data (a resolution of 128 Hz) are available only between 20:51 and 20:54 UT, thus, we calculate the propagating velocity of the hole-like structure between 20:51:55 and 20:52:56 UT based on timing analysis (Harvey, 1998) to verify whether these compressional structures are non-propagation. Figure 2A shows the positions of the MMS spacecraft relative to MMS1 at 20:52 UT. The inter-spacecraft distances are $\sim$13 to 21 km. Before performing the timing, the magnetic field data have been low-pass filtered with a cutoff period of 30 s to reduce the effect of high frequency fluctuations. Figure 2B shows the cross correlations between MMS1 and the three other satellites by using BZ. The maximum correlation coefficients are all almost 1 between MMS1 and MMS2/3/4 with a lag time of -0.312 s, -0.164 s and -0.039 s, respectively. The estimated velocity is (71.3, 11.7°, -28°) in spherical coordinates (r, $\theta$, $\varphi$) transferred from GSM coordinate system, where $\theta$ and $\varphi$ are the longitude and latitude, respectively. By contrast, the average ion velocity is (71.6, 37.8°, -28.4°) in this interval. Comparing these two velocities, one can find that the compressional structures in Figure 1 are approximately stationary, i.e. they are mirror mode structures. The determined velocity is the phase velocity in the spacecraft frame, i.e. the front of the structure is supposed to be perpendicular to the

determined velocity.

The minimum variance direction is supposed to be parallel to the above estimated velocity by timing, however, the angle between these two directions is ~37°. The MVA technique can be effected by waves or noises superimposed on the discontinuity surface, while the inter-spacecraft distances and configuration of the MMS satellites can effect on the accuracy of calculation, which might a possible explanation for the large difference between the two estimated normal directions.

We have added the above details in our revised manuscript.

(Harvey 1998) does not appear in the manuscript references list. I assume it is Chapter 12 in the ISSI "Analysis Methods for Multi Spacecraft Data" book.

Answer: Thanks for your nice comment. Yes, it is this reference. We have added the reference in our revised manuscript.

Page 6, line 127-135 The velocity used for estimating the scales (line 129) should be the one determined from timing analysis, not the plasma flow velocity. Since the two are not very different (line 118), this should not change much the results. Most probably the mirror mode structures have different sizes in different directions. For this study, the relevant size is the size in the direction orthogonal to the magnetic field. This size should be determined considering the angle between the mean magnetic field and the velocity vector determined from the timing analysis. Since the minimum variance direction – which should be close to the velocity direction – seems to be orthogonal to the mean magnetic field (figures 2 and 3), I expect that the sizes estimated in the manuscript are not far from the sizes in the orthogonal to the mean field direction. However, if the structures are not crossed through their centers – e.g. a path similar to the one shown in Figure 5 –, then the estimated sizes are only lower limits.

Answer: Thanks for your nice comment. Of course, it is better to use the velocity determined by timing to estimate the length scale of the mirror mode structure. The

inter-spacecraft distances are ∼13 to 21 km, which is too small to use the survey magnetic field data to do timing analysis. Only the burst magnetic field data during the first mirror mode structure are available, thus, we just do timing analysis for the first mirror mode structure to verify whether these structures are stationary in the ambient flow. Due to lack of sufficient burst magnetic field data, we estimate the length scale of the mirror mode structure in its cross-section using the M and N components of the ion velocity in our revised manuscript. It is difficult to verify whether the spacecraft trajectory crosses the center of the structure. Therefore, the estimated length is just the lower limits. We have added these details in our revised manuscript.

On lines 131-132 I assume the Authors meant "average ion perpendicular temperature".

Answer: Thanks for your comment. Yes, we meant "average ion perpendicular temperature". We have made a correction in our revised manuscript.

Page 7, Table 1 "i" should read "Scale (i)".

Answer: Thanks for your nice suggestion. In our paper, we mainly focus on the first and last mirror mode structures. And the information of these two structures have been written in the text. So, the table 1 is found to be not necessary to show, and has been deleted in our revised manuscript.

Page 7, lines 147-159 After line 147 the manuscript concentrates only on two magnetic dips (MM1 and MM5). To help readability, this should be clearly stated. The first structure (MM1) is analyzed in this paragraph and in the next one (up to line 181), while MM5 is analyzed in the remaining of the section. Dividing the text in subsections would improve readability. In this context, the maximum variance direction – which for magnetic mirrors should be aligned with the mean magnetic field – is the important direction. Therefore, the ratio between the maximum and the intermediate eigenvalues is relevant. The angles between the mean magnetic field and the determined L; M and N directions should be given. The current density should be computed also using the

curlometer, or the Authors should explain why this technique cannot be applied. Same comments apply for the MM5 on the next page.

Answer: Thanks for your nice comments and suggestions. We have separately analyzed these two mirror mode structures based on your suggestions. The angles between the mean magnetic field and the L, M and N directions are also given in the text. To study the relation between ions/electrons and the current density, the current density calculated by the curlometer method is a better choice. We determined the current density by the curlometer method, and did correlation analysis between the ion/electron velocity and the current density in our revised manuscript.

Figures 2 and 3 should show the orthogonal pressures of both ions and electrons. Are the ion velocities and the electron pressure in Figure 2 smoothed?

Answer: Thanks for your suggestions. We have shown the orthogonal pressures of both ions and electrons in these two figures. Only the electron data in these two figures have been smoothed within a 30-second window, since only electron data have significant high-frequency noise.

Page 7-8, lines 161-174 A more quantitative approach to determine which species (ions or electrons) contribute mostly to the electrical current is desirable. The Authors might e.g. compute the correlation between the electrical current and the ion and electron velocities.

Answer: Thanks for your nice suggestions. We have calculated the correlation coefficient between the electrical current and the ion/electron velocity in our revised manuscript. "The correlation coefficient between $jN$ and $VeN$ inside MM1 is -0.97." "The correlation coefficient between $ViN$ and $jN$ is 0.92 in the whole interval of MM2"

Page 11, lines 240-242 Please state the assumptions made for estimating the current density $jB$.

Answer: Thanks for your nice suggestion. BL changes $\sim$5 nT in MM1 between

20:52:30 and 20:52:56 UT, and half of the estimated length of MM1 is 2.05 × 103 km in the cross-section. Assuming that BM and BN are 0, and BL changes just along the trajectory of MMS, a current density jB with a value of ∼2 nA/m2 in the cross-section is necessary to be self-consistent with the magnetic field depression. We stated the assumption in our revised manuscript.

Page 11, lines 251-255 There is no reference to chaotic particles in (Constantinescu 2002). Perhaps the Authors refer to another paper?

Answer: Thanks for your comment. We have corrected the reference, which is Büchner and Zelenyi (1989).

Büchner, J., and Zelenyi, L. M. Regular and chaotic charged particle motion in magnetotail like field reversals. Journal of Geophysical Research, 94, 11,821–11,842. https://doi.org/10.1029/JA094iA09p11821, 1989.

Page 12-13, lines 285-295 An estimation of the gradient drift velocities for electrons and ions (similar with the estimation done in the previous paragraph for MM1), as well as an estimation of the electron diamagnetic drift should be given.

Answer: Thanks for your comments. We use the data in the time interval between 21:02:30 and 21:02:50 UT to estimate the ion thermal pressure and magnetic gradients. Also, the average ion perpendicular and parallel temperatures, average total magnetic field and average curvature radius in this interval are used to estimate the velocities of the ion drift motions. Consequently, the velocities of the ion diamagnetic, magnetic gradient and curvature drift motions are ∼17 km/s, 33 km/s and 79 km/s, respectively. By contrast, the velocities of the electron diamagnetic, magnetic gradient and curvature drifts are ∼5 km/s, 14 km/s and 36 km/s. Since the ion diamagnetic and magnetic curvature drifts move almost in the same direction in the M-N plane, while the ion magnetic gradient drift moves in the opposite direction. Thus, the collective drift velocity is ∼63 km/s, very close to the ion velocity inside MM2 with a speed of 70 km/s. Thus, one can expect that the bipolar ViN in Figure 4 is the collective behaviors of the

ion drift motions in MM2.

Page 13, lines 301-309 The normal directions (line 305) are almost orthogonal to each other. Knowing the estimated size between the entry and exit points, d, one can derive the transversal size of the structure as illustrated in Figure 5 (about 1:4d). Why is the MMS trajectory a curved line? Does the assumed relative motion of the magnetic structure change so much during the crossing time?

Answer: Thanks for your comments. "The normal directions are almost orthogonal to each other, the maximum length of MM2 in the cross-section could be 1.4 times the estimated length (6.6 i) based on the assumption of a circle." We found that the M component of the ion velocities ViM at two edges of MM2 are different, so the MMS trajectory was drawn as a curved line. Actually, the difference ViM at two edges of MM2 is not significant, so a straight line could be better to show the MMS trajectory. The trajectory has been changed to be a straight line in this figure in our revised manuscript.

Please also note the supplement to this comment:
https://www.ann-geophys-discuss.net/angeo-2019-144/angeo-2019-144-AC1-supplement.pdf

―――――――――――――――

---

## Author Comment (AC2) · 21 Jan 2020

General remarks Wang et al. investigated the roles of electrons and ions in the formation of the current in the mirror mode structures in the plasma sheet using by the MMS observations. They found that the electrons and ions play a different role in the different sizes of mirror mode structures: the current carriers are mainly the electrons in small size mirror mode by magnetic gradient-curvature drift, and the ions in large size mirror mode by the ion diamagnetic drift. This study sheds new light on formation of currents in the mirror modes, and is worthy of publication in AG after moderate reversion.

In the discussion section: MMS consists of four identical spacecraft, and could provide the simultaneous measurements of four points. Why the authors use the plasma measurements and magnetic field to estimate the time series of magnetic gradient curvature drift, electron diamagnetic drift, ion diamagnetic drift, and other terms. I think this is useful to estimate these different terms and then compare them.

Line 18 It would be better to replace "data" to "instruments", or remove "data".

Answer: Thanks for your nice suggestions. We have removed "data" from this sentence.

Line 47-49 Actually the sizes of magnetic holes can be less than ion cyclotron radius in the magnetosheath. Such magnetic holes, named as kinetic-size magnetic hole or electron vortex magnetic hole, are widely observed using by MMS (doi:10.3847/1538-4357/ab0f2f, doi:10.1002/2017JA024415, doi.org/10.3847/1538-4357/aac831, doi:10.1002/2016JA023858).

Answer: Thanks for your comments. "Magnetic holes with a scale less than 1 i also exist in the magnetosheath as well as in the plasma sheet, and electron vortices are found inside these kinetic-size structures (Huang et al., 2017, 2018, 2019; Yao et al., 2017)." We have added these sentences in our revised manuscript.

Line 54-56 The small-size magnetic holes, below one ion cyclotron radius, are also detected in the plasma sheet (doi.org/10.3847/1538-4357/ab0f2f). These magnetic holes are always accompanied with electron scale instabilities, such as whistler waves.

Answer: Thanks for your comments. "Magnetic holes with a scale less than 1 i also exist in the magnetosheath as well as in the plasma sheet, and electron vortices are found inside these kinetic-size structures (Huang et al., 2017, 2018, 2019; Yao et al., 2017)." "Mirror mode structures accompanied by electron dynamics and whistler waves are also reported to occur during the dipolarization processes (Li et al., 2014; Huang et al., 2018)." We have added these sentences in our introduction.

Line 61-63 Dipolarization fronts are widely investigate in many literatures

(doi:Âa10.1002/2015JA021083, doi:10.1029/2012GL051784, doi:10.5194/angeo-30-97-2012), and they play an important role in the energy conversion, mass transport, particle accelerations, and wave activities.

Answer: Thanks for your comments. "Dipolarization fronts (DFs), characterized by a sharp enhancement in BZ in GSM, are formed ahead of the earthward fast flows (Ge et al., 2012; Wu et al., 2013; Schmid et al., 2016; Xiao et al., 2017). They play an important role in the energy conversion, mass transport, particle accelerations and wave activities (Fu et al., 2012; Huang et al., 2012, 2015)." We have added these sentences in our introduction.

Line 72-75: I suggest the author give the motivation of this paper to help the readers to better understand their work.

Answer: Thanks for your nice suggestion. "Our aim is to figure out whether the main contributor to the current inside the ion-scale mirror mode structure is the electron or ion." We have added this sentence to the last paragraph of the introduction.

Line 80-83 (Russell et al., 2014) should be corrected to (Russell et al., 2016). If the author did not use the burst mode data in this paper, I suggest the authors remove the introduction about the resolution of burst mode in this part.

Answer: Thanks for your suggestions and comments. We have made correction to the reference. We have also deleted the description about the burst mode data in the introduction.

Line 115-117 Why are the data performed low-pass filtered before the timing analysis? I suggest the authors give some descriptions here.

Answer: Thanks for your nice suggestions. Waves with a period of several to ∼20 s can be found inside the magnetic dip in Figure 3. To reduce the effect of high frequency fluctuations, the data have been performed low-pass filtered before the timing analysis. We have given descriptions in our revised manuscript.

Line 120 "tends to be larger" "increases" or "has a peak"?

Answer: Thanks for your comments. We meant that the ion number density has a peak in the trough of the oscillations. To better descript the relation between the number density and the total magnetic field, we have changed this sentence to "The total magnetic field varies in anti-phase with the ion number density during this interval."

Line 131: how to calculate the local ion gyro radius? Which time interval? Please give the details in the text.

Answer: Thanks for your comments and suggestions. The local ion gyro radius is calculated by the average ion perpendicular temperature and the average magnetic field magnitude between 20:51:55 – 20:52:56 UT. We have given more details in our revised manuscript.

133-135 Why the range of the scale and the angle are inconsistent with those in table 1? The "rotation angle" is the "shear angle"?

Answer: Thanks for your comments. The angle is the angle between the magnetic field directions at two edges of each structure. The inconsistency here is caused by a typo. We have revised it in our manuscript. Since we mainly focus on the first and last mirror mode structures, and the information in table 1 can be found in the text, so table 1 is not necessary now and we have deleted it in our revised manuscript.

Line 155: As the authors know, the separation of the four MMS spacecraft is very small. Thus, one can use the curlometer method to estimate the current density based on the magnetic field from four spacecraft. Why not the author use this method to calculate the current and compare with the current derived from the plasma measurements?

Answer: Thanks for your comments and suggestions. The current density can be determined by the plasma moments or the curlometer method. To study the relation between ions/electrons and the current density, the current density calculated by the curlometer method is a better choice. So, we calculated the current by the curlometer

in our revised manuscript. And we compared the current with the electron/ion velocity, and found a strong relationship between the current and the electron or ion velocity.

Line 192 JN should be corrected to JM.

Answer: Thanks for your comments. We have revised it in our manuscript.

Line 216: "Ion velocities" should be "Ion velocities Vi_md"?

Answer: Thanks for your comments. We have revised it in our manuscript.

Line 230-235 The pileup region usually exists behind the DFs, not ahead of the DFs, for example the definition of flux pileup region in the paper (doi:10.1029/2012JA018141). In addition, the mirror mode structures are observed after the detection of DF. Why the authors thought they originate from the pileup region ahead of the DF?

Answer: Thanks for your suggestions. Yes, the flux pileup region usually occurs behind the DF, and the region ahead of the DF is called the pressure pileup region. I have changed "the pileup region" to "the pressure pileup region".

Since mirror mode structures are stationary in the ambient flow, we can estimate the distance of the structures relative to the DF in the Y direction using the average VY ∼30 km/s during the structures. Thus, they are likely to occur dawnside of the MMS spacecraft with a distance of ∼4 RE in the Y direction when the spacecraft are crossing the DF at around 20:38 UT. Compared this distance with the typical size of the DF (∼3 RE) (Huang et al., 2015) and the size of the structures, the mirror mode structures might come from the dawnside flank of the DF. Since the DF is considered to be a tangential discontinuity (Schmid et al., 2019) which pushes the background plasma to its flanks (Fu et al., 2012a, 2012b; Liu et al., 2013; Birn et al., 2015), the plasma near the flank is expected to come from the pressure pileup region ahead of DFs. In addition, mirror mode structures have been reported to be potentially generated in such a pressure pileup region (Zieger et al., 2011; Wang et al., 2016). Thus, the mirror mode structures in Figure 1 might originate from the pressure pileup region ahead of the DF.

Line 243 Please indicate at which time "The amplitude of the bipolar jN in MM1 is âĹij2 nA/m2".

Answer: Thanks for your suggestions. We have added the time interval in this sentence.

Line 247 It would be better to add "in MM5" after "electron velocity"

Answer: Thanks for your suggestion. We have revised it based on your suggestion.

Line 271-272 The authors can try to use the magnetic field from four MMS spacecraft to estimate the curvature radius of mirror modes.

Answer: Thanks for your suggestion. Using the data from all four MMS satellites, we can determine the curvature of MM1 by

d=Bˆ(-2) B_i âĹǦ_i B_j-Bˆ(-4) B_j B_i B_l âĹǦ_i B_l

where the indices i, j and l indicates the three components of the magnetic field, and B = |B| (Shen et al., 2003). The curvature radius RC is 1/d. We have estimated the curvature radius of the mirror mode structures, and discussed it in our revised manuscript.

Line 286-187 Did the authors ever calculate the magnetic gradient drift velocity in MM5? It would be necessary to compare the magnetic gradient drift velocity and the Vi_N.

Answer: Thanks for your suggestion. We use the data in the time interval between 21:02:30 and 21:02:50 UT to estimate the ion thermal pressure and magnetic gradients. Also, the average ion perpendicular and parallel temperatures, average total magnetic field and average curvature radius in this interval are used to estimate the velocities of the ion drift motions. Consequently, the velocities of the ion diamagnetic, magnetic gradient and curvature drift motions are ∼17 km/s, 33 km/s and 79 km/s, respectively. Since the ion diamagnetic and magnetic curvature drifts move almost in

the same direction in the M-N plane, while the ion magnetic gradient drift moves in the opposite direction. Thus, the collective drift velocity is ∼63 km/s, very close to the ion velocity inside MM2 (which is MM5 in our previous manuscript) with a speed of 70 km/s.

Line 304-307 Did the authors compare the normal directions calculated by MVA and timing method to ensure the accuracy of the results.

Answer: Thanks for your suggestion. Comparing the normal directions determined by MVA and timing method can ensure the accuracy of the results. However, no burst magnetic field data are available in this time interval. So, we did not compare the normal directions calculated by both methods in our manuscript.

Line 307-309 Please indicate which MMS? 1 or 2 or 3 or 4 after "trajectory of MMS". How to deduce the possible trajectory of MMS, please give details in the text.

Answer: Thanks for your comments and suggestions. Since the inter-spacecraft distances of MMS are very small compared to the scale of this mirror mode structure, the trajectory of the four MMS satellites are almost the same. Therefore, we only draw the trajectory of MMS1 in this figure. Based on the studies about the geometry of the mirror mode structure, we assume that the cross-section of the mirror mode structure is a circle. According to the normal directions of the both half sides of the structure and the ambient flow direction, we can simply get the possible trajectory of MMS1. We have added these details in our revised manuscript.

Line 316 Please add a color bar in Figure 5.

Answer: Thanks for your suggestion. We have added a color bar in this Figure.

Please also note the supplement to this comment:
https://www.ann-geophys-discuss.net/angeo-2019-144/angeo-2019-144-AC2-supplement.pdf